# DNA Methylation in Regulatory T Cell Differentiation and Function: Challenges and Opportunities

**DOI:** 10.3390/biom12091282

**Published:** 2022-09-12

**Authors:** Lu Bai, Xiaolei Hao, Julia Keith, Yongqiang Feng

**Affiliations:** Department of Immunology, St. Jude Children’s Research Hospital, 262 Danny Thomas Pl MS 351, Memphis, TN 38105, USA

**Keywords:** DNA methylation, epigenetics, regulatory T cells, DNMT1, DNMT3A, DNMT3B, TET1, TET2, TET3, Foxp3

## Abstract

As a bona fide epigenetic marker, DNA methylation has been linked to the differentiation and function of regulatory T (Treg) cells, a subset of CD4 T cells that play an essential role in maintaining immune homeostasis and suppressing autoimmunity and antitumor immune response. DNA methylation undergoes dynamic regulation involving maintenance of preexisting patterns, passive and active demethylation, and *de novo* methylation. Scattered evidence suggests that these processes control different stages of Treg cell lifespan ranging from lineage induction to cell fate maintenance, suppression of effector T cells and innate immune cells, and transdifferentiation. Despite significant progress, it remains to be fully explored how differential DNA methylation regulates Treg cell fate and immunological function. Here, we review recent progress and discuss the questions and challenges for further understanding the immunological roles and mechanisms of dynamic DNA methylation in controlling Treg cell differentiation and function. We also explore the opportunities that these processes offer to manipulate Treg cell suppressive function for therapeutic purposes by targeting DNA methylation.

## 1. Introduction

CD4^+^Foxp3^+^ regulatory T (Treg) cells play important roles in maintaining immune tolerance by providing dominant suppression over effector T and other immune cells. The majority of Treg cells develop in the thymus from CD4^+^CD8^−^ single positive (SP) thymocytes as a consequence of the recognition of self-antigens in the presence of interleukin (IL)-2 and are known as thymic Treg (tTreg) cells [1,2,3]. Treg cells can also be converted from naïve CD4^+^ T conventional (Tconv) cells in the periphery (pTreg) under favorable conditions, including antigen stimulation, IL-2, transforming growth factor β (TGF-β), and retinoic acid [4,5]. Despite different origins and target specificity, tTreg and pTreg cells appear to have comparable gene expression profiles and suppressive functions except for a few controversial molecular markers, such as Helios and Neuropilin-1 [6,7,8]. These Treg cells collectively confer immune tolerance toward a broad spectrum of antigens over an entire lifetime, ranging from self-antigens to those from food, commensal bacteria, tumors, and paternally-derived fetal antigens [2]. All Treg cells express nuclear protein Foxp3, which is essential for their identity, fitness, and immune-suppressive function regardless of the differentiation paths. Compared with Tconv cells, Treg cells also differentially express a number of other proteins involved in signal transduction, transcriptional regulation, and immune-suppressive activity, such as cytotoxic T-lymphocyte associated protein 4 (CTLA-4), high-affinity IL-2 receptor α-chain (CD25), glucocorticoid-induced TNFR-related protein (GITR), inducible T cell costimulator (ICOS), transcription factors Helios (Ikzf2) and Eos (Ikzf4), CD73, transforming growth factor-β (TGF-β), and IL-10 [2]. Treg cells are dynamically regulated by tissue environmental cues in a context-dependent manner reflected by activation and/or differentiation to subpopulations that express additional transcription factors (e.g., Gata3, T-bet, PPAR-γ, and RORγt) and signaling molecules (e.g., CXCR4, ST2, and CD103) [9,10,11,12,13]. This multilayered gene regulation in differentiated Treg cells enables adaptable control of immune tolerance according to tissue microenvironments.

Among factors dictating Treg cell differentiation and function, Foxp3 has been the most intensively investigated. Here, we focus on the transcriptional regulation of Foxp3 expression, as it is a key step in determining a gene’s function. Foxp3 is known to be regulated by four distinct conserved noncoding sequences (CNS) or enhancers in addition to a conserved promoter (Figure 1A) [14,15,16]. Genetic deletion of individual CNS elements indicates that they play non-overlapping roles in dictating Foxp3 induction or maintenance during Treg cell development or after lineage commitment, respectively. CNS0 is regulated by histone methyltransferase Mll4 and Satb1 and is required for efficient Foxp3 induction in response to IL-2 signaling (Figure 1B) [15,17,18]. CNS1 serves as a platform for Smad molecules to facilitate Treg cell induction in a TGF-β-dependent manner, which appears to be important for preferentially maintaining immune tolerance at mucosal surfaces [19,20,21]. Like CNS0, CNS3 is poised by the monomethylation of histone H3 at lysine 4 (H3K4me1) during hematopoiesis and promotes Treg cell generation by increasing the sensitivity of precursor cells to TCR signal strength in the presence of other induction cues [22,23]. These different mechanisms thus expand Treg cell repertoires against a broad spectrum of targets to confer robust immune tolerance. 

In contrast, CNS2 is dispensable for Foxp3 induction and Treg cell development but exclusively required for maintaining heritable Foxp3 expression in committed Treg cells (Figure 1B). This region is especially rich in CpG sites that are fully methylated in Treg precursor and other cell types [24,25]. In mature Treg cells, these CpG sites are completely demethylated, forming a Treg-specific demethylated region (TSDR), presumably contributing to stable Foxp3 expression, Treg cell fate, and durable suppressive function. Likewise, TSDRs at other loci, such as *Il2ra*, *Ctla4*, *Ikzf2*, and *Ikzf4*, may contribute to optimal Treg cell function [14,16,26,27,28]. Therefore, the differentiation and function of Treg cells are governed by both genetic and epigenetic programs interwoven during stepwise induction and lineage maintenance [29], among which differential DNA methylation appears to play unique roles. 

DNA methylation is the conversion of cytosine (C) to 5-methylcytosine (5mC), which occurs primarily at CpG dinucleotides in mammals [30]. The methylation of CpG sites generally represses gene expression by preventing the binding of transcriptional activators (transactivators) or by increasing the association of transcriptional repressors. DNA methyltransferases (DNMTs) are specialized enzymes responsible for DNA methylation. Though emerging evidence shows that there is further complexity [31] in this process, a simplified view is that DNMT1 copies DNA methylation patterns from parent strands onto newly synthesized strands during replication [32,33], whereas DNMT3A and DNMT3B mediate *de novo* DNA methylation. DNMT3L is catalytically inactive and regulates DNMT3A/DNMT3B activity [34]. However, because it is not expressed in T cells, we do not further discuss it in this review. In the case of Treg cells, inhibition of DNA methylation or promotion of passive DNA demethylation enhances the transcriptional induction or maintenance of Foxp3 [35,36], indicating the important role of DNA methylation in Treg cell fate determination. Ten-eleven translocation (TET) enzymes are responsible for DNA demethylation via active conversion of methylated CpG (mCpG) to unmethylated CpG in a multi-step enzymatic reaction [27]. DNA demethylation is generally regarded as a permissive condition to upregulate gene expression by increasing chromatin accessibility, binding of transactivators, and/or release of transcriptional repressors [37]. Genetic evidence indicates that TET-mediated epigenetic regulation is essential for maintaining Foxp3 expression and durable Treg suppressive function [27,29,38,39]. 

Here, we review recent research progress on DNA methylation involved in Treg cell development, lineage stability and plasticity, as well as immune suppressive function. We discuss the questions and challenges for further investigating the immunological roles and underlying mechanisms of DNA methylation in regulating these processes. We also explore the opportunities for manipulating Treg cell suppressive function for therapeutic purposes by targeting DNA methylation. 

## 2. Maintenance of DNA Methylation Patterns by DNMT1 Restricts Treg Cell Induction

### 2.1. Maintenance of DNA Methylation Patterns

During mitosis, DNA replicates via a semiconservative mechanism and is then split into two daughter cells. Preexisting DNA methylation patterns are copied to newly synthesized DNA strands, conferring an inheritable epigenetic state that maintains the stability of cellular function (Figure 2A). This process relies on DNMT1, which has a high affinity for hemi-methylated DNA and methylates newly synthesized CpG sites during DNA replication [40]. Because CpG sites are short palindromic sequences, this mechanism enables DNMT1 to precisely copy preexisting DNA methylation patterns from parent to nascent strands. 

DNMT1 contains several functional domains, including a C-terminal catalytic domain and N-terminal regulatory domains [41]. In particular, the N-terminal independently folded domain (NTD) binds to multiple proteins, including proliferating cell nuclear antigen (PCNA), an essential component of DNA replication forks. DNMT1 is self-inhibited in resting cells. During DNA replication, E3 ubiquitin ligase UHRF1 (ubiquitin-like with PHD and RING finger domains 1) recruits DNMT1 to hemi-methylated DNA through the replication foci-targeting domain (RFT) of DNMT1 (Figure 2A). This activates DNMT1 to methylate newly synthesized DNA strands according to parental methylation patterns. Thus, DNA semiconservative replication coupled with DNMT1 precisely maintains DNA methylation patterns during cell division. Blockage of this process results in cell-cycle-dependent dilution of DNA methylation, causing DNA hypomethylation and various dysregulations of cell differentiation or function depending on cell type [42]. 

### 2.2. DNMT1 in Thymic T Cell Differentiation

Ablation of *Dnmt1* by *Lck^Cre^* (via insertion of a modified Cre downstream of the *lck* proximal promotor) [43] in experimental mice starting from the CD4^−^CD8^−^ double negative (DN) stage of thymocyte development results in approximately 90% loss of CD4^+^CD8^+^ double positive (DP), CD4^+^CD8^−^, and CD4^−^CD8^+^ SP thymocytes (Figure 2B) [44]. This severe defect is most likely caused by reduced cell survival. The number of T cells in the peripheral lymphoid organs also significantly decreases due to reduced viability [44]. It is also conceivable that the real effects may be substantially underestimated because of the incomplete deletion of *Dnmt1*. As impaired cell survival is dominant, the full function of Dnmt1 and maintenance of DNA methylation in early thymocyte differentiation are unclear. The exact mechanism causing cell death by *Dnmt1* deletion and DNA hypomethylation is not well understood. The profound consequence on DP thymocyte survival immediately following *Dnmt1* deletion by *Lck^Cre^* at the DN stage suggests that maintenance of preexisting DNA methylation patterns is crucial during early thymocyte differentiation involving extensive cell proliferation. 

Deletion of *Dnmt1* by *Cd4^Cre^* (expression of Cre is controlled by the *Cd4* gene promoter, enhancer, and silencer) [44] starting at the DP stage results in comparable numbers of CD4^+^ SP and CD8^+^ SP T cells in the thymus and peripheral lymphoid organs [44,45]. This might be due to SP cells’ lower proliferation rate or decreased sensitivity to *Dnmt1* loss compared with DP cells. However, *Dnmt1* deficiency impairs T cell activation and proliferation in vivo, consistent with the role of Dnmt1 in maintaining DNA methylation patterns during DNA replication and cell division. Interestingly, although thymic CD4^+^Foxp3^+^ Treg frequency appears to be normal in *Dnmt1*-deficient CD4^+^ SP thymocytes, *Dnmt1* deficiency in CD8^+^ SP thymocytes markedly increases Foxp3^+^ cells [45], suggesting that DNA methylation normally restricts Foxp3 expression in CD8^+^ T cells. More remarkably, non-Treg CD4^+^ and CD8^+^ SP thymocytes and peripheral naïve CD4^+^ T cells isolated from *Dnmt1*-deficient (by *Cd4^Cre^*) mice readily upregulate Foxp3 expression upon TCR stimulation even in the absence of IL-2 and TGF-β, both known to be required for Foxp3 induction in wild-type counterparts [45]. As Foxp3 induction in *Dnmt1*-deficient T cells is cell division dependent [45], unchanged Foxp3^+^ cell number among *Dnmt1*-deficient CD4^+^ SP thymocytes is probably due to limited divisions from DP to SP stages. In contrast, Foxp3-expressing cells are increased among CD8^+^ SP thymocytes, most probably because some CD8^+^ T cells undergo additional proliferation compared with CD4^+^ SP thymocytes [46]. 

Mechanistically, TGF-β signaling was shown to induce Uhrf1 nuclear exclusion and degradation, which results in the inhibition of DNMT1 activity and subsequently causes DNA hypomethylation after cell division and Foxp3 induction upon TCR stimulation [47]. This model provides an interesting epigenetic basis for TGF-β-dependent Treg cell development. However, unlike *Dnmt1*-deficient T cells, *Uhrf1*-deficient T cells are viable and proliferate after stimulation. It is unknown whether DNA hypomethylation induced by TGF-β via Uhrf1 is genome-wide or target specific. If this effect applies to all mCpG sites, it would be important to uncover the mechanism causing different phenotypes of *Dnmt1* and *Uhrf1* deficiencies in T cells. If DNA hypomethylation only occurs in particular regions, including *Foxp3 cis*-regulatory elements, how the target specificity is achieved warrants further investigation. 

When taken together, these experiments suggest that a preexisting, unspecified DNA methylation pattern opposes Foxp3 induction and Treg cell development. This epigenetic modification restricts Treg lineage commitment to predominantly CD4^+^ T cells upon exposure to induction cues, thus helping to distinguish from other guidance signals that dictate the differentiation of alternative T cell subtypes. 

### 2.3. DNMT1 Is Essential for Differentiated Treg Cells

*Lck^Cre^*- and *Cd4^Cre^*-induced ablation of *Dnmt1* affects both Tconv and Treg cells that constantly communicate during immune homeostasis. To understand the cell-intrinsic role of Dnmt1 in Treg cells, Treg-specific deletion of *Dnmt1* would be more informative. Given the nearly exclusive expression of Foxp3 by Treg cells, Cre line driven by the endogenous *Foxp3* gene (i.e., *Foxp3^Cre^*) [48] induces conditional deletion of *Dnmt1* in only differentiated Treg cells. This results in lethal systemic autoimmunity and an extremely shortened lifespan comparable to those of *Foxp3^null^* or scurfy mice (Figure 2B) [49,50,51], suggesting that Treg-dependent immune tolerance is nearly completely destroyed. This dysregulation is likely caused by both markedly reduced Treg cell number (quantity) and suppressive capacity (quality) [49]. *Dnmt1*-deficient peripheral Treg cells have a shortened lifespan due to increased apoptosis upon TCR stimulation. Because Treg cells undergo continuous activation and self-renewal through TCR and IL-2 signal-dependent proliferation [52,53,54,55], the critical role of Dnmt1 in Treg suppressive function further proves the significance of maintaining preexisting DNA methylation patterns during DNA replication and cell division. However, as discussed above, the exact mechanism by which DNA hypomethylation in the absence of *Dnmt1* causes cell death is unknown. In addition, the dominant, severe effect of *Dnmt1* deficiency on Treg cell survival may mask other functions of DNA methylation that remain to be determined. Unbiased methods, such as acute deletion of *Dnmt1* and cell division tracing, could be used to achieve this goal. In particular, it may prove to be very important to specify whether maintenance of DNA methylation patterns by Dnmt1 in differentiated Treg cells is also crucial for sustaining other key features of Treg cells, such as lineage stability, expression of immune suppressive molecules, and restriction of effector T cell programs. Future experiments are also required to pinpoint the genetic elements controlled by DNA methylation to regulate the genes governing these processes.

Similar to *Dnmt1* deletion, Treg-specific deletion of *Uhrf1*, which determines Dnmt1 chromatin binding, results in lethal autoimmune inflammation [56]. In a separate study, deletion of *Uhrf1* by *Cd4^Cre^* generates a proliferative defect in colonic Treg cells [57]. Overall, these studies reveal distinct roles of Dnmt1 in controlling Treg cell development and survival, indicating context-dependent functions of maintaining DNA methylation patterns in Treg precursor cells and differentiated Treg cells. 

### 2.4. DNMT1 in De Novo DNA Methylation

In addition to the widely accepted function in maintaining DNA methylation patterns, Dnmt1 was recently shown to play a significant role in *de novo* DNA methylation in mouse embryos [31,58]. This result raises a question about the significance of maintaining DNA methylation demonstrated by *Dnmt1* deficiency. A careful examination of the cell division dependency would be able to address this question. The study of Dnmt1 *de novo* methylation activity may help to elucidate locus-specific regulation of DNA methylation by Dnmt1 in T cells, which may be comparable to *de novo* methylation mediated by Dnmt3a and Dnmt3b (discussed below). For example, GITR in human T cells, which is highly expressed and plays an important role in Treg cell function [59], appears to be directly targeted and repressed by Dnmt1 and mCpG-binding domain protein 4 (MBD4) [60]. In addition, there is evidence that the transcriptional levels of Dnmt1 vary significantly during murine T cell differentiation and activation (www.immgen.org (accessed on 10 September 2022)), suggesting that Dnmt1 activity may be dose dependent. Although its significance and mechanism are unknown, maintenance of DNA methylation patterns might not be uniformly conducted across the genome but rather regulated by available Dnmt1. Alternatively, Dnmt1 expressed above the basal level that is required for maintaining DNA methylation might contribute to *de novo* DNA methylation by interacting with nuclear proteins bound at specific genetic elements. Distinguishing these two models would produce important new insights into Treg cell development and function controlled by Dnmt1-dependent DNA methylation. 

### 2.5. Regulation of DNMT1 Function

In addition to transcriptional regulation, Dnmt1 also undergoes posttranslational modifications, including phosphorylation, acetylation, methylation, and ubiquitylation [61,62]. However, the exact effects of these modifications on the Dnmt1 function remain to be determined. In particular, protein phosphatase 6 (*Pp*6)-deficiency results in increased CpG methylation of *Foxp3* enhancer CNS2, which is linked to impaired Dnmt1 dephosphorylation, reduced Foxp3 expression in Treg cells, and enhanced Akt signaling [63]. Future experiments are required to establish the causal roles of Dnmt1 phosphorylation in DNA methylation of CNS2 and Treg lineage stability and function. In addition, Dnmt1 and Uhrf1 expression are also dynamically regulated. For example, IL-6 increases DNMT1 expression and activity in human erythroleukemia cells [64], while Uhrf1 is upregulated in colonic Treg cells upon colonization of commensal bacteria [57]. These results suggest context-dependent regulation of *de novo* DNA methylation or maintenance of DNA methylation by Dnmt1. 

## 3. *De Novo* DNA Methylation by DNMT3 in Reprogramming Treg Cells

### 3.1. DNMT3A and DNMT3B Structures

DNA methylation of new sites or at higher levels that are not inherited from parental cells is established by *de novo* methylation. This process is traditionally thought to be catalyzed by DNMT3 family enzymes (i.e., DNMT3A and DNMT3B; Figure 3A) [65,66,67]. DNMT3A and DNMT3B are homologous and have similar domain structures. Like DNMT1, the catalytic domains of DNMT3A and DNMT3B are located at the C-terminals. There are two domains on the N-terminals, namely PWWP and ADD domains. The former mediates the association with chromatin, and the latter forms an autoinhibitory loop by interacting with the catalytic domain [68,69,70]. Chromatin binding disrupts the interactions between ADD domains and catalytic domains, thus releasing the autoinhibitory loop to enable DNA methylation. This model suggests that the targeting specificity of DNMT3A and DNMT3B and, thereafter, *de novo* DNA methylation are determined by the chromatin-binding proteins associated with DNMT3A and DNMT3B [70]. 

### 3.2. Effects of DNMT3A or DNMT3B Deficiencies

Germline deficiencies of *Dnmt3a* or *Dnmt3b* cause runted mice or developmental defects at late embryonic stages, respectively. These defects reveal essential, non-overlapping roles of Dnmt3a and Dnmt3b in embryonic development, consistent with their distinct expression patterns in embryos [32]. However, deficiency of both *Dnmt3a* and *Dnmt3b* leads to embryonic lethality, indicating functional redundancy, overlapping roles, or synthetic genetic interactions of these two enzymes. In mature T cells, Dnmt3a is expressed at a significantly higher level than Dnmt3b (www.immgen.org (accessed on 10 September 2022)), suggesting that Dnmt3a is predominant. 

Conditional deletion of *Dnmt3a* by *Cd4^Cre^* does not cause obvious dysregulation of Treg cells but results in DNA hypomethylation of the *cis*-regulatory elements of *Ifng* and *Il4* and subsequent misexpression of IFNγ and IL-4 by Tconv cells under conditions that normally restrict the production of these cytokines in wild-type Tconv cells [71]. It has been well established that helper T cell lineages, e.g., Th1, Th2, and Th17 cells that produce distinct effector molecules, are induced by discrete environmental cues to express unique lineage-determining factors such as T-bet, Gata3, or RORγt [72]. The dysregulation of IFNγ and IL-4 expression in *Dnmt3a*-deficient Tconv cells suggests that Dnmt3a-dependent *de novo* DNA methylation plays an important role in restricting the boundaries among helper T cell lineages. Future experiments are required to determine how Dnmt3a is selectively targeted to these loci to repress gene expression and ensure distinct helper T cell lineage identities. Furthermore, a revelation of the extent to which misexpressed effector molecules in *Dnmt3a*-deficient Tconv cells affect immune response would produce insights into the biochemical nature of rigid T cell lineages and their immunological functions. 

### 3.3. Potential Roles of DNMT3A in Treg Cells

Although no obvious dysregulation of Treg cell development or function was observed in the resting state upon deletion of *Dnmt3a* by *Cd4^Cre^* [71], Dnmt3a might play a critical role in Treg cells under particular contexts that require reprogramming of Treg cell function (Figure 3B). Treg cells have been shown to be incredibly versatile, reflected by further differentiation into distinct subtypes upon exposure to tissue environmental cues or loss of Treg cell identity to become “exTreg” cells under adverse conditions [3,73,74]. In the former case, the Treg cell function is modulated by additional programs, such as the expression of transcription factors Gata3, T-bet, RORγt, Bcl-6, or PPARγ in different contexts [75]. Whole genome DNA methylation profiling of Treg cells isolated from skin, liver, adipose tissue, or lymph nodes reveals profound differential DNA methylation at selected genetic elements [76]. As DNA hypermethylation is contributed by *de novo* DNA methylation, Dnmt3a might be steered by environmental cues to enable Treg cell adaptation to specific tissues, thus generating tissue-specific epigenetic memories and fine-tuning Treg cell function. 

In the latter case, adverse conditions (e.g., pro-inflammatory cytokines, TCR stimulation, and cell proliferation) drive the transdifferentiation of Treg cells to exTreg cells characterized by silenced Foxp3 expression and production of effector molecules (Figure 3B) [77,78,79,80,81,82,83]. During this transition, the CpG sites at the *Foxp3* promoter and CNS2, two TSDRs, are remethylated to a level comparable to that of CD4^+^ Tconv cells [24]. Inhibition of DNA methylation by 5-aza-2′-deoxycytidine stabilizes Foxp3 expression in natural Treg cells and prevents exTreg generation in vitro, suggesting that Dnmt3a-dependent *de novo* DNA methylation is responsible for silencing Foxp3 transcription. Consistent with this notion, SUMO E3 ligase PIAS1 was shown to recruit Dnmt3a to the *Foxp3* promoter to restrict Foxp3 expression [84]. This model also suggests an interesting mechanism for fine-tuning Treg suppressive function where Dnmt3a confers lineage plasticity to Treg cells to favor effector T cell programs and promote inflammation. Future experiments are required to test this possibility and establish the causal role of Dnmt3a and DNA *de novo* methylation in Treg cell transdifferentiation. 

These two potential roles of Dnmt3a in Treg cells appear to be paradoxical; the former (i.e., adaptation to tissue environments) appends additional features to the core Treg suppressive program, and the latter (i.e., transdifferentiation to exTreg cells) subtracts Treg function to promote inflammation. These opposite outcomes can be reconciled by target-specific activities of Dnmt3a linked to different environmental cues so that some drive Treg adaptation to gain additional functions, whereas others destabilize Treg cell fate and induce transdifferentiation of Treg cells to effector-like exTreg cells. Given the factors and pathways responding to extracellular stimulations, Dnmt3a-dependent *de novo* methylation is likely one of the many mechanisms governing Treg adaption or transdifferentiation. This leads to the question of whether Dnmt3a plays indispensable or predominant roles among these mechanisms. Nonetheless, Dnmt3a-dependent DNA *de novo* methylation could be a double-edged sword in regulating Treg cell function and reprogramming, which warrants further scrutiny to determine the targeting mechanisms of Dnmt3a and their immunological roles. 

### 3.4. Transdifferentiation of Treg Cells

Whereas the potential role of Dnmt3a in Treg cell adaptation to tissue microenvironments requires experimental evidence, extensive research has been performed on Treg cell transdifferentiation to effector-like exTreg cells. We previously showed that Dnmt3a coprecipitates with STAT6 protein, which may lead to the methylation of CpG sites at the *Foxp3* promoter and CNS2 and silencing of Foxp3 expression after Treg cell exposure to IL-4 (Figure 3B) [24]. Interestingly, inhibition of cell division blocks Foxp3 silencing in adverse environments in vitro, indicating that loss of Foxp3 transcription depends on the cell cycle. Thus, cell proliferation creates a vulnerable period enabling cell fate reprogramming. As Treg cell proliferation is a result of combined stimulations by TCR, co-receptors, and cytokines (e.g., IL-2, IFNγ, IL-4, or IL-6) [2], this mechanism allows Treg cells to adjust suppressive capacity in inflammatory conditions by inducing transdifferentiation to exTreg cells through an apparently stochastic process during cell division [24]. 

Although the mechanistic nature of this transition is unclear and requires further investigation, cell division appears to serve as a timer together with signals derived from adverse environmental cues, including inflammatory cytokines, to reset the fate of a fraction of Treg cells. In this scenario, Dnmt3a-mediated *de novo* DNA methylation, though it can be achieved in a cell-cycle-independent manner, might be limited to a vulnerable period that appears during Treg cell division in the presence of inflammatory cytokine signaling that targets Dnmt3a to the *Foxp3* locus as well as other TSDRs to restore effector T cell programs. It is unknown whether *de novo* DNA methylation in this setting is an outcome of the over-competition of Dnmt3a against Tet enzymes that catalyze active DNA demethylation (discussed below). Here, Tet enzymes are driven by the signaling pathways that favor Treg cell fate, which is continuously responsible for the maintenance of hypomethylated TSDRs. This mechanism may be readily outcompeted by Dnmt3a, which is targeted by inflammatory cytokine signaling, during Treg cell division in a stochastic manner. This model is supported by increased mCpG levels at TSDRs, unstable Foxp3 expression, and derepressed effector molecules in *Tet2*/*Tet3*-deficient Treg cells induced by *Foxp3^Cre^* [38]. 

Transdifferentiation of Treg cells to effector-like T cells has been linked to a variety of inflammatory diseases [85], but the associated DNA methylation has not been rigorously investigated. Key questions need to be addressed to support this model. First, proof of Treg cell transdifferentiation in these pathological settings that rely on stringent genetic tracing systems is needed. Two Cre-loxp reporter tools have been generated to achieve this goal. One is constitutive *Foxp3^Cre^* knock-in mice; however, a major caveat exists for this model. These mice have been shown to be able to mislabel a significant fraction of non-Treg cells in the presence of Rosa-loxP-Stop-loxP (*Rosa^LSL^*) fluorescence protein reporters due to promiscuous expression of Foxp3 during Treg cell development [86,87,88]. A more robust cell tracing system is based on inducible *Foxp3^CreER^* knock-in mice that present an expression pattern nearly identical to endogenous Foxp3 after acute tamoxifen treatment when combined with *Rosa^LSL^* reporter mice [89]. Because of the high fidelity, *Foxp3^CreER^* mice have been widely used to label Treg cells and trace their fate or to perform acute deletion or overexpression in the presence of *Rosa^LSL^* reporter, loxP-flanked conditional alleles, or *Rosa^LSL^-*linked-genes of interest, respectively. To trace Treg cell fate, *Foxp3^CreER^ Rosa^LSL^* fluorescence reporter mice are acutely treated with tamoxifen followed by a period of rest or immune challenge, if necessary. Labeled T cells are then isolated from lymphoid and non-lymphoid organs to examine the expression of Foxp3 and effector molecules. *Foxp3^CreER^ Rosa^LSL^* mice offer a reliable approach to examining Treg cell lineage stability and transdifferentiation in the steady state or in disease settings with or without additional perturbations. 

Second, the immunological significance of Treg cell transdifferentiation to effector-like exTreg cells in physiological and pathological environments needs to be assessed with full consideration of the side effects of experimental manipulations. For example, cell sorting, in vitro manipulations, and adoptive transfer of Treg cells to lymphopenic recipients during routine suppression assays often affect Treg cell fitness, TCR diversity, and other unspecified features. On the other side, protein factors, signaling pathways, or metabolic regulators controlling Treg lineage stability also modulate Treg immune suppressive function, which limits the manipulation of Treg lineage stability without side effects [2,85]. Thus, unambiguous determination of the biological role of Treg cell transdifferentiation appears to be a technical challenge. 

*Foxp3* enhancer CNS2 could potentially offer a solution because it acts in *cis* and is almost exclusively required for the maintenance of Foxp3 expression and not for Foxp3 induction or other genes’ expression [23,24,25]. CNS2 deficiency impairs Foxp3 inheritable expression during Treg cell activation and division when IL-2 is deprived or in the presence of pro-inflammatory cytokines. This results in chronic tissue inflammation in aged mice or after immune challenges, such as chronic viral infection or immunization by self-antigens causing experimental autoimmune encephalomyelitis [24,25]. Further impairment of the stability of Foxp3 expression in Treg cells by a combined deficiency of CNS0 and CNS2 causes early onset lethal autoimmune disease comparable to that in *Foxp3^null^* mice [14]. Together, these results prove that the maintenance of Foxp3 expression or Treg lineage stability is crucial for immune homeostasis and suppression of autoimmune diseases. However, it is unknown to what extent exTreg cells contribute to autoimmune inflammation in these mice or play a role under specific immunological contexts. Furthermore, the functional significance of exTreg cells has not been determined in untouched animals without isolation and adoptive transfer of Treg or exTreg cells into lymphopenic animals. 

### 3.5. Distinct Roles of DNMT3A and DNMT3B

Although DNMT3B is expressed at lower levels in Treg cells compared with DNMT3A, it may still play an important role. This possibility is supported by the observation that Dnmt3b binds to a methylated *Foxp3* upstream enhancer in Tconv cells but not in Treg cells [90], suggesting that Dnmt3b also regulates DNA methylation of the *Foxp3* locus. Further study is needed to know whether Dnmt3a and Dnmt3b have overlapping functions in Treg cells, which can be assessed by comparing the effects of single and double deficiencies of *Dnmt3a* and *Dnmt3b*. 

There is also evidence that Dnmt1 contributes to *de novo* DNA methylation (discussed above) [31,58]. Given that the N-terminal domains of Dnmt1, Dnmt3a, and Dnmt3b are not well conserved and potentially associate with proteins bearing different DNA sequence-binding specificities, these enzymes, to a large degree, likely play non-overlapping roles by methylating the CpG sites of distinct target genes through their unique recruitment mechanisms. In order to test this notion, future experiments are required to reveal the interacting proteins and chromatin targeting mechanisms of Dnmt1, Dnmt3a, and Dnmt3b in Treg cells and to uncover the effects on Treg cell differentiation or transdifferentiation after conditionally deleting *Dnmt* genes individually or in combination.

### 3.6. Genes Regulated by De Novo DNA Methylation

CpG sites of many Treg function-related genes (e.g., *Tcf7* and *Bcl2*) are hypermethylated in Treg cells isolated from skin and adipose tissues compared with those in Treg cells from lymphoid organs [76]. Thus, *de novo* DNA methylation may modulate the expression of genes required for Treg cell adaptation to tissue microenvironments. Likewise, transdifferentiation of Treg cells to exTreg cells may regain DNA methylation at numerous TSDRs, including *Foxp3* promoter and enhancer CNS2 [24]. To fully understand the molecular nature and immunological function of the latter process, differential DNA methylation levels across the genome need to be first characterized by comparing Treg and exTreg cells isolated *ex vivo* with whole genome bisulfite sequencing. Next, acute ablation of *Dnmt3a* and *Dnmt3b* alone or together in Treg cells can be used to uncover the effects of *de novo* DNA methylation on gene expression in Treg cells or exTreg cells by tracing Treg cell fate with inducible *Foxp3^CreER^* and Rosa reporter mice. Third, because Foxp3 as a lineage-determining factor regulates the expression of hundreds of genes in Treg cells, this effect should be separated from the role of *de novo* DNA methylation in controlling other gene expressions by ablating Foxp3 alone or together with Dnmt proteins. Finally, future experiments are also needed to assess the interaction between Foxp3 and Dnmt proteins to determine their epistasis in regulating target gene expression. 

### 3.7. Mechanisms of DNA Methylation in Regulating Gene Expression

Although the overall roles of Dnmt enzymes and *de novo* DNA methylation in Treg cells can be assessed by genetic ablation experiments, how differential DNA methylation controls gene expression in a locus-specific manner remains unclear. In vitro binding assays have documented a list of proteins whose DNA binding is directly regulated by DNA methylation [91,92]. For example, transcriptional activators Runx1 and Cbf-β binding at *Foxp3* enhancer CNS2 are inhibited by DNA methylation in vitro [23]. This result, in principle, suggests a mechanism by which DNA methylation of CNS2 and other *cis*-regulatory elements inactivates Foxp3 transcription by excluding the binding of transactivators. Methylated DNA may also recruit transcriptional repressors, directly or indirectly, to actively suppress transcription [93]. It remains to be tested whether and how this mechanism could mediate the silencing of Foxp3 transcription. Notably, mCpG binding proteins have also been shown to play other roles. For example, deletion of *Mbd2* in mice causes hypermethylation of CNS2, most likely due to impaired Tet1 and Tet2 binding, and dampens thymic Treg cell development and immune suppressive function [94]. In another case, deletion of *Mecp2* by *Foxp3^Cre^* impairs Treg cell function without affecting DNA methylation levels at CNS2, leading to elevated autoimmune inflammation in a mouse model of transfer colitis [95]. Thus, the exact functions of methylation-sensitive DNA binding proteins should be examined experimentally with a consideration of their direct effects on the expression of Foxp3 and other Treg function-related genes. 

In addition, Dnmt enzymes may regulate gene expression through their associated proteins, which might be independent of their enzymatic activities [96,97]. This possibility can be tested by replacing wild-type Dnmts with catalytically inactive mutants in the endogenous loci and then assessing Treg cell differentiation and function. How *de novo* DNA methylation precisely regulates target gene expression in a locus-specific manner requires examination of the associated proteins and their dependency on DNA methylation status. 

## 4. DNA Demethylation in Maintaining Treg Cell Fate and Function

### 4.1. Passive DNA Demethylation Due to Impaired DNMT1 Function

DNA methylation levels are dynamically regulated during the differentiation and function of Tconv cells and Treg cells [29,76]. As described previously, the reverse reaction of DNA methylation can be achieved passively via dilution of methylated DNA during DNA replication and cell proliferation when DNMT1 function is impaired globally or locally (Figure 4A) [98,99]. This mechanism explains the effects of *Dnmt1* conditional deletion on Tconv cells and Treg cells discussed above, where the failure to maintain DNA methylation patterns results in DNA hypomethylation. A study of Uhrf1 in TGF-β-dependent Foxp3 induction suggests that passive DNA demethylation is precisely controlled in T cells [47]. However, the extent and target specificity of passive DNA demethylation, as well as their roles, remain to be fully explored [96]. Further testing of the notion of targeted passive DNA demethylation can be achieved by examining the factors interacting with DNMT1 and regulating its target-specific function during T cell division. It is also possible that transcription factors may directly compete with DNMT1 for DNA binding, disrupting DNA methylation and leading to hypomethylation after DNA replication and cell division. In order to reveal the gene loci selectively controlled by passive DNA demethylation in T cells, whole genome bisulfite sequencing can be used to profile differential DNA methylation in the absence of active DNA demethylation (below) before and after T cell proliferation. 

### 4.2. Active DNA Demethylation by TET Enzymes

Active DNA demethylation is catalyzed by TET enzymes through a multi-step reaction starting from the hydroxylation of 5-methylcytosine (5hmC) followed by 5-formylcytosine (5fC) and 5-carboxylcytosine (5caC) (Figure 4A) [37,100]. Three homologous enzymes, TET1, TET2, and TET3, have been identified in vertebrates, and all contain a C-terminal catalytic domain sequentially oxidizing 5mC, 5hmC, and 5fC in the presence of Fe^2+^, α-ketoglutarate (α-KG), and vitamin C or ascorbic acid (Figure 4B) [101,102]. TET1 and TET3 have a conserved CXXC domain on the N-terminal to bind DNA. In contrast, TET2 DNA-binding capability is conferred, at least partially, by the CXXC domain of IDAX protein (also named CXXC4) that interacts with TET2 [102,103]. This mechanism may play a critical role in recruiting TET enzymes to DNA and determining their target specificities. However, to what extent the DNA-binding specificities of TET proteins are determined by the CXXC domains is unclear. It is also possible that additional DNA-binding proteins may complex with TETs to selectively demethylate certain gene loci, which is further discussed below. 

TET enzymes alone cannot demethylate DNA by converting mCpG to CpG. Instead, the oxidized forms, 5fC and 5caC, can be recognized by thymine DNA glycosylase (TDG) and replaced with unmodified cytosines by the base-excision repair (BER) system (Figure 4A) [104,105,106]. Alternatively, 5hmC, 5fC, or 5caC can be diluted during DNA replication if DNMT1 fails to recognize these modified cytosines (discussed above) (Figure 4A) [107]. The exact contributions of TDG/BER-mediated base replacement versus passive dilution of oxidized cytosines to TET-dependent DNA demethylation are unclear. A recent study showed that whereas *Tdg* germline deficiency is embryonically lethal coupled with hypermethylation of CpG islands [108], acute deletion of *Tdg* by CreER and tamoxifen does not affect Treg cell induction and Foxp3 expression [109]. Because TET enzymes are required for stable Foxp3 expression and Treg suppressive function [38,39], the subtle effect, if any, of acute *Tdg* deletion suggests that TDG may play a minor role in TET-induced DNA demethylation in T cells. It will be important to know whether TDG/BER-mediated base replacement or passive demethylation following TET-dependent oxidization of methylcytosine is executed constitutively or contributes differentially to DNA demethylation in a context-dependent manner that creates more precise regulation. Regardless of these downstream processes, overall, TETs initiate this multi-step enzymatic reaction to induce target-specific active DNA demethylation, enabling a new layer of gene regulation. 

### 4.3. Active DNA Demethylation Controls Treg Lineage Stability

Profound DNA hypomethylation at TSDRs was observed in differentiated Treg cells compared with Tconv cells long before the discovery of TET-dependent active demethylation [26,28]. Accumulating evidence now suggests that active DNA demethylation plays a critical role in maintaining Treg cell fate and suppressive function [26,35,56,110,111]. At the *Foxp3* locus, enhancers CNS1 and CNS2 are fully demethylated upon Treg lineage commitment. Conditional deletion of *Tet2* and *Tet3* in Treg precursor cells by *Cd4^Cre^* proves that this process relies on TET enzymes [27]. Germline deletion of CNS2 does not significantly affect Treg cell development in the thymus or periphery. Instead, CNS2-deficient Treg cells have a marked impairment of stable Foxp3 expression during Treg cell division when IL-2 is deprived or in pro-inflammatory conditions [24,25]. This leads to reduced Treg suppressive function manifested by mild inflammation in aged mice, after chronic viral infection, or during metabolic dysregulation. These phenotypes are consistent with *Tet2/Tet3*-deficient Treg cells upon *Cd4^Cre^*-induced deletion [27]. However, the latter causes a more profound impairment of stable Foxp3 expression in Treg cells, presumably because CNS2 demethylation only contributes partially to TET-dependent stabilization of Foxp3 transcription. 

In contrast, CNS1 appears to be only required for Treg cell induction, which precedes DNA demethylation [19,20,21]. It is unknown what role CNS1 demethylation plays in Treg cells. One possibility is that TET-dependent DNA demethylation of multiple genetic elements, including CNS1 and CNS2, generates new regulatory circuits that could play redundant roles in maintaining Foxp3 transcription in committed Treg cells. In this model, CNS1′s function could be dominated by other mechanisms, such as CNS2-mediated transcriptional regulation. Thus, in the absence of CNS1, other regulatory circuits might be sufficient to sustain Foxp3 expression without causing significant impairment in experimental settings that lack the capability to fully reveal the biological significance. In another scenario, TET enzymes may induce broader DNA demethylation around their chromatin anchoring positions (e.g., CNS2), which is comparable to other epigenetic modifications, such as histone acetylation H3K27ac and methylation H3K4me3 that are extensively distributed beyond the enzyme binding sites [112]. Future experiments are required to distinguish these two models to bring insights into the nature of TET function and the mechanisms regulating Treg lineage stability. 

### 4.4. TET Enzymes Are Required to Maintain Unmethylated DNA and Treg Cell Fate

Because CNS1 and CNS2 are nearly completely demethylated in committed Treg cells [27,29], one would anticipate that unmethylated DNA is autonomously inherited during cell proliferation in the steady state. Surprisingly, conditional deletion of *Tet2* and *Tet3* by *Foxp3^Cre^* results in a significant reestablishment of DNA methylation at TSDRs, including CNS1 and CNS2 [38,39]. This result indicates that maintenance of unmethylated CpG sites in TSDRs relies on continuous TET function, likely by opposing *de novo* DNA methylation catalyzed by DNMT3A/DNMT3B. Because mice bearing *Tet2*/*Tet3* dual deficient Treg cells have a significantly longer lifespan than CNS0/CNS2 double deficient mice (further discussed below) [14,38,39], both affecting Treg lineage stability (Figure 4C), *de novo* DNA methylation and transdifferentiation of Treg cells in the absence of *Tet2* and *Tet3* appear to be controlled by a slow process. TET-mediated active DNA demethylation and DNMT3-dependent *de novo* DNA methylation represent opposing programs and likely constantly compete at TSDRs to dictate Treg cell fate and immune suppressive function (Figure 5). Given that Treg cells are continuously induced from precursor cells and transdifferentiate to exTreg cells in *Tet2/Tet3* conditionally deficient mice, the disease onset and methylation levels of TSDRs may not accurately exhibit the kinetics of *de novo* DNA methylation and Treg fate loss. Thus, future experiments are required to measure the kinetics of DNA remethylation of TSDRs and Treg cell fate change upon acute ablation of Tet enzymes with or without blockade of basal inflammatory cytokines, such as IL-4, TNFα, IL-6, and IFNγ. In addition, accurate quantification of the competition and equilibrium of chromatin binding of TET and DNMT3 proteins will also produce insights into this important process (further discussed below). 

### 4.5. Treg Lineage Stability Is Controlled by More Than DNA Demethylation

To fully understand the mechanisms controlling Treg lineage stability, recently, we systematically searched for additional genetic elements around the *Foxp3* locus that coordinate with CNS2 to maintain Foxp3 expression. This led to the identification of CNS0, which acts together with CNS2 to confer stable Foxp3 expression (Figure 4C) [14]. In the absence of both CNS0 and CNS2, Treg cells abruptly silence Foxp3 expression upon adoptive transfer to T cell-deficient mice, an assay widely used to assess Treg lineage stability [24]. This result indicates significant redundancy between CNS0 and CNS2 in maintaining Foxp3 expression, although CNS0 deficiency alone only impairs Foxp3 induction [15,16]. Interestingly, CNS0 serves as a STAT5-binding site in response to IL-2 signaling, an essential component of Treg induction cues [1,2,3]. CNS0 is also regulated by histone methyltransferase Mll4 and nuclear structure protein Satb1 (Figure 1B) [17,18]. However, all these roles appear to be independent of TET function because the CpG sites in CNS0 are free of DNA methylation in CD4^+^ Tconv cells and Treg cells [14]. Thus, the stability of Foxp3 expression relies on both DNA demethylation-dependent and –independent mechanisms. The redundancy between CNS0 and CNS2 confers robust Foxp3 expression and Treg lineage identity for durable immune tolerance under various adverse environments; in the absence of either mechanism, Foxp3 expression can still be largely sustained. 

Enhancer redundancy appears to be a general mechanism conferring the functional robustness of gene expression, cell differentiation, or organogenesis [113,114,115]. Thus, other redundant mechanisms might also have evolved to reinforce the stability of Foxp3 expression. For example, we and others showed that the coordination between CNS0 and CNS3 plays a role in maintaining Foxp3 expression, although single deficiencies of either CNS appear to only decrease Foxp3 induction [14,16]. Among these mechanisms, TET-dependent DNA demethylation plays a predominant role, which cannot be compensated for by other mechanisms in the absence of TETs or CNS2. Protein factors associated with these genetic elements provide clues for understanding the underlying mechanisms. For example, STAT5 and Foxp3 bind to CNS2 only after DNA demethylation during Treg cell development, the former responding to IL-2 signaling and the latter forming a feed-forward regulatory loop [29]. Surprisingly, the stability of Foxp3 transcription is considerably resistant to IL-2 deprivation or Foxp3 deletion [29], suggesting that CNS2 demethylation establishes multiple regulatory circuits acting in parallel to maintain Foxp3 transcription such that perturbations of individual pathways produce minimal effects [14,16,29]. Thus, this mechanism embeds another layer of Foxp3 transcriptional control. Unlike CNS2, the binding of STAT5, Satb1, and Mll4 to CNS0 is not controlled by TET enzymes or DNA methylation [15,16,17,18]. Likewise, Satb1, Mll4, and c-Rel (downstream of NF-kB signaling) may also bind to CNS3 in a DNA methylation-independent manner [17,18,23]. In addition, several other factors (such as Smarcd1, Brd9, and Usp22) have recently been shown to act on these genetic elements to stabilize Foxp3 expression through potentially separate pathways [29,116,117]. Therefore, it is likely that additional factors and pathways which are yet to be discovered also play important roles in maintaining Foxp3 expression.

When taken together, this intricate regulation supports the overall model that stable Foxp3 expression and thereafter durable Treg cell fate are sustained by multiple DNA methylation-dependent and -independent mechanisms that respond to a variety of extracellular cues and intracellular states. Regulation of Treg cell fate by multiple inputs offers numerous switches to reprogram Treg cells and adjust immune tolerance. Future experiments are expected to delineate these mechanisms and reveal their dual roles in controlling Treg cell fate flexibility under physiological and pathological conditions. 

### 4.6. Active DNA Demethylation Controls Treg Suppressive Function

Apart from *Foxp3* enhancers that undergo drastic active DNA demethylation, a number of Treg-function-conferring genes are also controlled by Treg-specific DNA demethylation, such as *Tnfsf18*, *Ctla4*, *Ikzf4*, and *Il2ra* [28]. Interestingly, almost the same levels of DNA demethylation were observed at these loci in “wannabe” Treg cells that report Foxp3 transcription but lack functional Foxp3 protein [28]. This is consistent with the notion that TET enzymes are recruited to TSDRs by an upstream pathway to demethylate these CpG sites, which is independent of Foxp3 expression. TCR signaling appears to be required for this process [28]; however, it alone is insufficient to determine the target specificity because DNA demethylation is not induced by TCR stimulation at TSDRs in CD4 Tconv cells. Therefore, Treg induction cues, including TCR, IL-2, and TGF-β signaling, act together to determine the site specificity and/or activity of TET enzymes such that TSDRs are demethylated at various levels [27,28,29,76]. Upon deletion of *Tet2* and *Tet3* by *Foxp3^Cre^*, Treg cells—before losing Foxp3 expression—drastically elevate the expression of many genes controlling cell proliferation and effector T cell function, including those associated with T follicular helper (Tfh) and Th17 cells [38,39]. Because the expression of effector molecules is highly suppressed in wild-type Treg cells, another avenue of investigation concerns how effector T cell programs are derepressed when *Tet2* and *Tet3* are ablated in the presence of Foxp3. Identification of the TSDRs whose hypermethylation in the absence of *Tet2* and *Tet3* disrupts Treg immune suppressive programs will produce valuable insights. Intriguingly, because DNA hypermethylation normally suppresses gene expression, it is possible that the downregulation of certain proteins by hypermethylated TSDRs indirectly activates effector T cell programs in Treg cells before Foxp3 expression is lost. Future experiments are expected to test this model and uncover the mechanisms governing Treg reprogramming. 

### 4.7. Tet-Deficient exTreg Cells Are Refractory to Treg Suppression

Remarkably, conditional deletion of *Tet2* and *Tet3* by *Foxp3^Cre^* in Treg cells results in autoimmune inflammation, which cannot be suppressed by wild-type Treg cells [38]. This raises a fundamental question about the mechanisms of Treg-mediated immune suppression acting on both Treg and Tconv cell sides. One possibility is that exTreg cells derived from *Tet2*/*Tet3*-deficient Treg cells are refractory to Treg suppression, probably because these cells inherit high-affinity TCRs to self-antigens [118]. This TCR inheritance would confer a competitive advantage over wild-type Treg cells. However, this model is not applicable to CNS0/CNS2 double deficient Treg cells that have a significantly impaired lineage stability [14]; in this case, exTreg cells, if any, are fully controlled by wild-type Treg cells in heterozygous females or mixed bone marrow chimeric mice because these animals do not develop any noticeable autoimmunity [14]. Therefore, *Tet2*/*Tet3*-deficient Treg cells or exTreg cells are probably reprogrammed to a pro-inflammatory state and are resistant to the suppressive function of wild-type Treg. This model is supported by enhanced expression of cell cycle genes and effector molecules in *Tet2*/*Tet3*-deficient Treg cells [38]. In addition, TET-dependent DNA demethylation appears to be continuously required for Treg suppressive function via several distinct mechanisms beyond Foxp3 expression. Uncovering the underpinnings of this striking phenomenon of *Tet2*/*Tet3*-deficient Treg cells will provide important clues revealing the fundamental mechanisms of Treg suppression and the conditions in which these mechanisms are disrupted in autoimmune diseases. 

### 4.8. Metabolic Control of TET Activity

TET enzymatic activity relies on several co-factors, including α-KG, Fe^2+^, and vitamin C (Figure 4B) [119]. Vitamin C is a micronutrient and serves as an electron donor to reduce and recycle oxidized iron in the catalytic domains of TET enzymes during the reaction [120,121]. Vitamin C treatment activates TET enzymes to demethylate CNS1 and CNS2, stabilizing human and murine in vitro induced Treg (iTreg) cells, suggesting a crucial role of vitamin C in Treg cell development and function in vivo [27,29,122]. On the other side, oxidization of methylcytosine by TET enzymes is coupled with the conversion of α-KG to succinate and CO_2_ [123]. Because α-KG is an intermediate of the tricarboxylic acid (TCA) cycle, TET-dependent DNA demethylation is tightly linked to a cell’s metabolic state, and enzymes regulating the aerobic respiration cycle potentially modulate TET function (Figure 4B). 

This notion is supported by several lines of evidence. First, 2-hydroxyglutarate (2-HG), an analog of α-KG, acts as a competitive inhibitor of α-KG-dependent enzymes, including TETs [124]. It is converted from α-KG by mutant isocitrate dehydrogenase (IDH1 and IDH2) that often occurs in glioma and acute myeloid leukemia (AML) patients (Figure 4B) [125]. As a result, many CpG sites are hypermethylated in AML patients bearing IDH1/2 mutations [126]. Blocking 2-HG production by aminooxy acetic acid was shown to facilitate the demethylation of the *Foxp3* promoter and CNS2, leading to enhanced Foxp3 expression in Th17 cells [127]. Second, loss-of-function mutations of succinate dehydrogenase (SDH), which catalyzes the conversion of succinate to fumarate in the TCA cycle (Figure 4B), in a gastrointestinal stromal tumor, pheochromocytoma, and paraganglioma lead to dysfunctional TCA cycle and DNA hypermethylation, suggesting impaired TET function [128]. Third, the administration of glucose, glutamine, or glutamate, which feed into the TCA cycle, increases the level of 5hmC along with α-KG in the liver of the recipients [129]. 

Apart from metabolites, hydrogen sulfide was also shown to promote the demethylation of the *Foxp3* promoter and CNS2 by maintaining TET1 and TET2 expression via transcription factor NFYB upon sulfhydration [130]. This observation raises a general question about the extent to which TET enzymes function in a concentration-dependent manner, either due to the stoichiometry of TETs and substrates or due to TET enzymatic activity. 

When taken together, these results suggest an intriguing model in which metabolic dysregulation may reprogram Treg cells via TET-dependent DNA demethylation, which could have a long-lasting effect on immune tolerance. 

### 4.9. Targeting Mechanisms of TET Enzymes

How TET enzymes trigger DNA demethylation at specific genomic regions is unclear. A working model states that proteins with DNA binding capabilities target TET enzymes to particular regions and demethylate the CpG sites. This model is supported by the observation that TGF-β and IL-2 signaling facilitate TET1 and TET2 binding to the *Foxp3* locus, perhaps via phosphorylated Smad3 and STAT5, respectively [130,131]. However, these two pathways are apparently insufficient to determine the target specificity of TET proteins at the *Foxp3* locus and other TSDRs. We propose a complex model to solve this mystery where multiple signaling pathways and nuclear proteins converge to guide TET chromatin binding and/or enzymatic activity (further discussed below). 

## 5. Future Directions

Dynamic DNA methylation catalyzed by DNMT and TET enzymes has been demonstrated to play crucial roles in Treg cell differentiation, lineage stability, immune suppressive function, and transdifferentiation. Despite significant progress, many important questions remain to be fully addressed. In addition to what has been discussed above, here we highlight a few major areas that require extensive exploration.

### 5.1. Targeting Mechanisms of DNMT and TET Enzymes

Whereas CpG sites subject to *de novo* methylation or active DNA demethylation can be accurately mapped by whole genome bisulfite sequencing of *Dnmt*- or *Tet*-deficient Treg cells, how the target specificities are determined is poorly understood. We propose that DNMT and TET enzymes are targeted via multivalent protein–protein interactions resulting from the convergence of signaling pathways and chromatin-associated proteins. 

In order to test this model, future experiments are required to first reveal the protein complexes of DNMTs and TETs in Treg cells during their lineage commitment or transdifferentiation. This can be achieved by routine purification of protein complexes of DNMT or TET enzymes followed by proteomic analysis [132]. Epitope-tagged DNMTs or TETs expressed at the endogenous levels, which can be obtained by inserting epitope tags into the *DNMT* or *TET* loci, would be preferred to maintain the fidelity of protein complexes and to take advantage of high-quality epitope antibodies. To capture weak and transient protein–protein interactions, proximity tagging coupled with “shotgun” proteomics can be used to identify proteins located within the proximity of bait proteins [133]. Additional methods will then be needed to distinguish between direct and indirect interactions. Next, mapping of chromatin binding and perturbation of these factors alone or in combination with genetic deletions (such as CRISPR knockout) can be employed to assess their roles in determining target-specific DNA methylation or demethylation by DNMTs or TETs in Treg cells.

Overall, the integration of biochemical and genetic approaches will be key to addressing the mechanisms by which DNMT or TET enzymes are targeted to specific *cis*-regulatory elements to regulate DNA methylation levels and gene expression, thus dictating Treg cell differentiation, lineage stability, and reprogramming.

### 5.2. Overlapping and Non-Overlapping Functions of DNMT or TET Enzymes

Diversification of the *DNMT* and *TET* genes, including *cis*-regulatory elements and protein amino acid sequences, may have evolutionarily created distinct expression patterns and protein functions [134] among these enzyme families. It is unclear to what extent DNMT and TET proteins play non-overlapping roles. This issue can be addressed by the conditional deletions of individual *DNMT* or *TET* genes in precursor cells or Treg cells, followed by the examination of DNA methylation patterns and Treg cell development, lineage stability, transdifferentiation, and immune suppressive functions. For example, TET2 appears to play a unique role in controlling cytokine gene expression (e.g., *Ifnγ* and *Il17*) in Tconv cells [127,131]. A study of *Tet2* deletion alone in Treg cells might reveal that it affects Treg cell lineage stability and/or function. Our recent results indicate that Treg cells harbor a considerable buffering capacity for robust immune suppressive function such that a mild impairment in Treg cell development or lineage stability is largely tolerated [14,22]. Therefore, this phenotypic robustness should be quantified, and Treg cell function assessed comprehensively during the investigation of individual DNMT or TET enzymes. Studies of the non-redundant functions of *DNMT* or *TET* genes would not only produce valuable insights into the diversification of *DNMT* and *TET* genes but also shed light on the regulatory space of Treg-mediated immune tolerance. 

As discussed in previous sections, several lines of evidence indicate that DNMT and TET enzymes, respectively, play significantly overlapping roles [27,32,38]. Two non-mutually exclusive mechanisms should be considered for the interpretation of these results: (a) DNMTs and TETs are truly redundant in methylating or demethylating the same DNA substrates or genomic loci; (b) genes are regulated by DNMT and TET enzymes via additive effects at non-overlapping genomic regions. In both cases, conditional ablation of all *DNMT* or *TET* enzymes causes more severe effects on Treg cells compared with single deletions. However, the former mechanism confers the robustness of DNA-methylation-dependent gene regulation, which is resistant to the perturbations of one of the enzymes [135], whereas the latter belongs to non-specific synthetic genetic interactions that apply to any genes [136]. In-depth analysis of DNA methylation levels across the genome and gene expression changes in Treg cells after conditional deletions of single or multiple *DNMT* or *TET* genes would be able to distinguish these two distinct mechanisms. 

Non-overlapping and over-lapping roles of DNMT and TET enzymes also suggest a gradient of DNA methylation at the genome scale due to different enzymatic activities. We propose that graded DNA methylation controls Treg cell differentiation, lineage stability, immune suppressive function, or transdifferentiation in a dose-dependent manner and under dynamic regulation by DNMT and TET enzymes (further discussed below). This notion can be tested by examining differential DNA methylation and Treg cell behaviors when *DNMT3* or *TET* genes are deleted partially or completely. Overall, these studies will produce important insights into the mechanisms and immunological roles of dynamic DNA methylation.

### 5.3. Mechanisms of Differential DNA Methylation in Regulating Gene Expression

In principle, DNA methylation controls gene expression by regulating the affinity of protein binding to DNA [91,92]. This creates a huge regulatory space, given the large number of genes controlled by dynamic DNA methylation [76]. DNMT and TET enzymes dictate the expression of hundreds of genes via different targeting mechanisms. Future comprehensive analyses are required to enrich this general model from several different angles. First, factors whose DNA binding is modulated by differential DNA methylation and their roles in DNA-methylation-dependent regulation of gene expression in Treg cells remain to be fully characterized. Second, DNA methylation and demethylation may trigger sequential reactions at chromatin, including but not limited to protein association and dissociation, histone modifications, chromatin architecture, and accessibility changes, as well as alterations of chromatin looping. These effects can be assessed by available epigenetic tools and computational algorithms. Third, extensive functional studies are needed to fully reveal the roles of graded DNA methylation and demethylation (discussed above) in Treg cells. For example, previous research showed that Foxp3, Treg function-related genes, and cell cycle and cytokine genes are regulated by TET-dependent DNA demethylation [38,39]. However, the functional significance of these changes and the underlying mechanisms are unclear. Further exploration of the differentially expressed genes in *Dnmt3* or *Tet*-deficient Treg cells would produce an in-depth understanding of the features conferred on Treg cells by *de novo* DNA methylation or active DNA demethylation.

### 5.4. Dynamic Regulation of DNA Methylation by the Competition between DNMT and TET Enzymes

DNMT and TET enzymes have been shown to compete and cooperate to regulate gene expression [137,138]. In the *Foxp3* locus, DNMT3 and TET enzymes appear to continuously compete in fully committed Treg cells because conditional ablation of *Tet2*/*Tet3* by *Foxp3^Cre^* results in the spontaneous remethylation of CNS1 and CNS2 coupled with unstable Foxp3 transcription in the steady state [38,39], suggesting that Foxp3 transcription and thereafter Treg cell fate are constantly targeted by opposing pathways acting on DNMT and TET enzymes. This competition is likely determined by environmental cues and intracellular states that favor Treg or effector T cell programs (Figure 5). As reprogramming of Treg cell fate requires cell division that relies on TCR and cytokine stimulation [24], this dynamic regulation of Treg cell fate in principle enables precise fine-tuning of immune tolerance in an antigen- and environment-specific manner, avoiding systematically dampening immune suppression. This mechanism would promote inflammation during infection or lead to focused autoimmunity if particular self-antigen stimulation impairs Treg lineage stability in the presence of adverse cues.

Future experiments are required to determine how this dynamic process is precisely controlled. First, it is crucial to reveal the protein factors or pathways targeting DNMT and TET enzymes to TSDRs in Treg cells, as discussed in previous sections. Second, studies are required to shed light on the equilibrium of *de novo* methylation and demethylation of TSDRs in response to different environmental cues (e.g., anti- or pro-inflammatory cytokines) before and after acute ablation of *Dnmt* or *Tet* enzymes. This would help to understand how the equilibrium of DNA methylation and demethylation is established and maintained and how this balance shifts during Treg differentiation, maintenance, and transdifferentiation upon exposure to different stimulations. In order to reveal the entire spectrum of dynamic DNA methylation, locus-specific equilibria of DNA methylation and demethylation should be measured and compared.

### 5.5. Targeting DNMT and TET Enzymes to Modulate Treg Cells

Selective dampening or enhancement of Treg cell function has been proposed to treat cancer or autoimmune diseases, respectively [139]. Targeting DNA methylation or demethylation pathways offers a unique approach to modulating Treg-mediated immune tolerance. It provides several advantages over other approaches because DNA-methylation-dependent epigenetic memory, once induced, is relatively long-lasting compared with transient stimulations that normally do not cause epigenetic changes or establish a sustainable gene regulatory network [24,29,30,38,122,140]. Targeting DNA methylation in certain conditions may provide such specificity and efficacy that cannot be readily achieved by other methods. For example, DNA demethylation has been shown as a key switch determining both Treg cell induction and lineage stability by facilitating the expression of Foxp3 and a number of Treg-function-conferring genes that are predominant over many other genes that are also regulated by DNA demethylation. If accomplished during Treg cell development, unmethylated DNA “locks in” Foxp3 transcription and Treg cell fate, conferring remarkable resistance to the perturbations of environmental cues [27,29]. Given these features, targeting DNMTs and TETs that function at the transition stages of Treg fate determination or reprogramming would be an effective method to modulate Treg-dependent immune tolerance. This goal can be achieved from the angles discussed in the following sections.

#### 5.5.1. Targeting DNMT and TET Enzymes

Several inhibitors of DNMT and TET enzymes have been discovered [141,142,143]; however, their efficacy appears to be very limited. Future research is needed to develop more potent compounds with enhanced specificity and reduced toxicity. Due to the complexity of Treg cell function in different disease settings, conditions to modulate DNMT and TET enzymes should be tested individually. Given the pleiotropic roles of differential DNA methylation in many other cell types, strategies to directly target DNMTs and TETs by pharmaceutical inhibitors remain to be fully explored to selectively modulate Treg cells in vitro or in vivo for therapeutic purposes.

#### 5.5.2. Targeting the Recruitment Mechanisms of DNMT and TET Enzymes

Understanding the chromatin targeting mechanisms of DNMT and TET enzymes could lead to other methods to modulate Treg cell differentiation and function. A number of signaling pathways have been targeted to pharmacologically modulate Treg cell development or function [139]. Future research is expected to determine the extent to which these pathways act on *de novo* DNA methylation or active demethylation to regulate Treg cells. In addition, as discussed previously, target specific *de novo* DNA methylation and active DNA demethylation appear to be determined not by single pathways but rather by the convergence of multiple pathways. The revelation of the targeting mechanisms of DNMTs and TETs would help to design new methods to modulate Treg cells by optimizing the effects on DNA methylation or demethylation at specific stages of Treg differentiation or reprogramming.

#### 5.5.3. Targeting Specific Gene Loci

Recently, genome editing technologies have created unprecedented opportunities to directly modify the epigenome to achieve target-specific gene regulation [144]. For example, fusing catalytically inactive Cas9 (dCas9) with DNMT3 or TET enzymes enables the editing of DNA methylation levels at specific gene loci via single guide RNA (sgRNA) molecules. As a proof of concept, this method has been successfully used to deliver the dCas9-Tet1 catalytic domain (CD) to the *MyoD* locus to activate its expression [145]. However, targeting dCas9-Tet1CD to *Foxp3* CNS2 only induced partial DNA hypomethylation and a minor, if any, stabilization of murine iTreg cells [146,147]. These results suggest that the efficiency of epigenome editing may depend on genomic contexts. Future experiments are needed to uncover the limiting factors of epigenome editing and to develop methods to solve this issue. Integration of epigenome editing with other approaches discussed above might be more effective. However, a potential side effect of this strategy is that dCas9 fusion proteins may block transcription factor binding, thus interfering with gene expression. Inducible or transient epigenome editing may be able to overcome this limitation.

#### 5.5.4. Targeting the Co-Factors of DNMT and TET Enzymes

As reviewed in previous sections, TET-dependent DNA demethylation requires vitamin C, α-KG, and Fe^2+^ [148]. Their availabilities control TET activities, leading to enhanced or decreased DNA demethylation. In addition, one-carbon metabolism regulates the levels of S-adenosyl methionine (SAM), the methyl donor for DNA methylation catalyzed by DNMT enzymes [149]. Pathways regulating SAM levels would directly control the enzymatic activities of DNMTs. These co-factors can be directly or indirectly targeted via different approaches, such as through metabolites administered as food supplements or via intravenous infusion, to selectively modulate DNMT or TET enzymatic activities, thus affecting Treg cell development, lineage stability, or reprogramming. This treatment can also be combined with other methods discussed above to more effectively manipulate DNA methylation levels in Treg cells.

## 6. Conclusions

As a bona fide epigenetic marker, DNA methylation is dynamically regulated by DNMT and TET enzymes in a gene locus- and cell-type-specific manner. Maintenance of DNA methylation patterns, *de novo* methylation, and active DNA demethylation were shown to play crucial roles in Treg cell differentiation, lineage maintenance, immune suppressive function, and transdifferentiation. This progress triggers further questions about the biological contexts, cell-extrinsic and intrinsic regulators, molecular mechanisms controlling differential DNA methylation, and their roles at different stages of the Treg lifespan. Future investigations are expected to address these fundamental questions and explore novel methods to modulate Treg cells by targeting DNMT and TET enzymes for the treatment of immunological diseases and cancer.

## Figures and Tables

**Figure 1 biomolecules-12-01282-f001:**
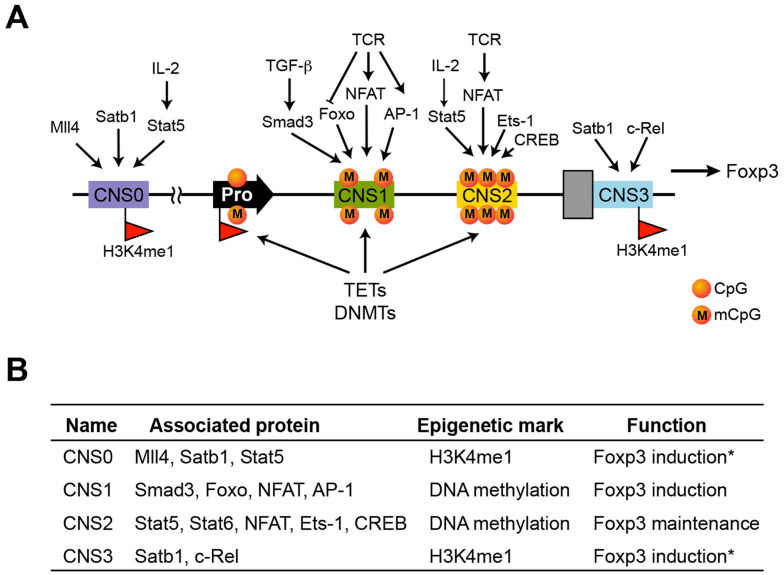
*Cis*-regulatory elements controlling Foxp3 expression. (**A**) Schematic of the *cis*-regulatory elements of the Foxp3 gene. Representative binding proteins and their upstream pathways are shown. CNS0 is localized in an intron of upstream gene *Ppp1r3f* (not shown). Pro, promoter. (**B**) Summary of *Foxp3* enhancers and their representative binding proteins, epigenetic modifications, and roles in Foxp3 expression. * CNS0 and CNS3 were recently shown to play a role in maintaining Foxp3 expression in the absence of CNS2 or CNS0, respectively [14,16].

**Figure 2 biomolecules-12-01282-f002:**
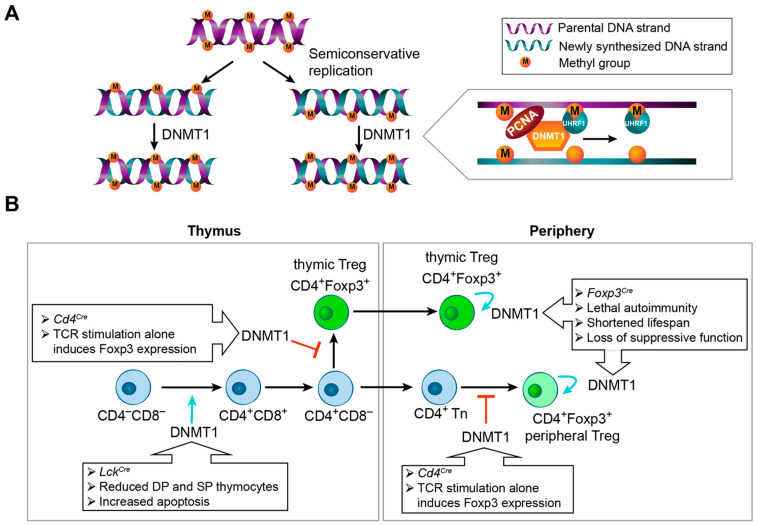
DNMT1 and the maintenance of DNA methylation patterns in Treg cell lineage specification and function. (**A**) Schematic of DNMT1 in maintaining preexisting DNA methylation patterns during semiconservative DNA replication and cell division. DNMT1 associates with PCNA and UHRF1 to bind to hemi-methylated DNA at the replication fork to methylate newly synthesized CpG sites. Impaired DNMT1 function would lead to DNA hypomethylation upon cell division. (**B**) Key effects of the conditional deletions of *Dnmt1* by *Lck^Cre^*, *Cd4^Cre^*, or *Foxp3^Cre^* on the differentiation of precursor cells and Treg cells in mice.

**Figure 3 biomolecules-12-01282-f003:**
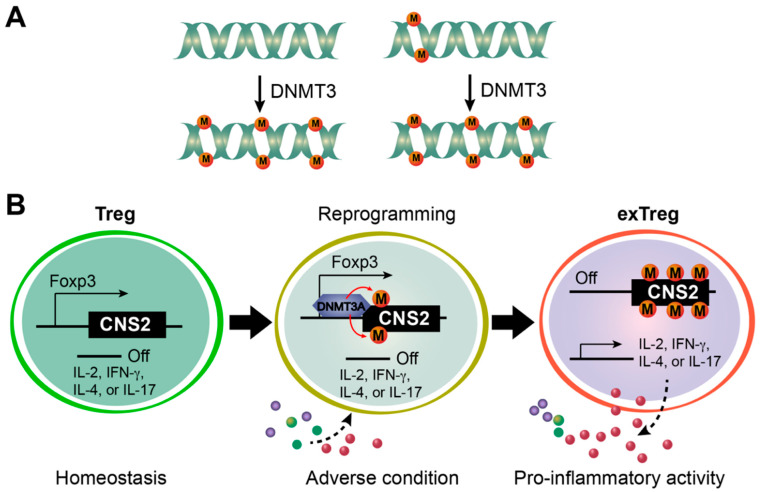
*De novo* DNA methylation in Treg cell transdifferentiation and reprogramming. (**A**) Schematic of *de novo* DNA methylation by DNMT3 (i.e., DNMT3A or DNMT3B), including methylation of new CpG sites and increased methylation levels of previous CpG sites at the population level. This active process is cell cycle independent. (**B**) A hypothetical model of *de novo* DNA methylation in Treg cell transdifferentiation and reprogramming. In adverse conditions, pro-inflammatory cytokine signaling drives DNMT3A to methylate the CpG sites at TSDRs, leading to the transdifferentiation of Treg cells to effector-like exTreg cells characterized by silenced Foxp3 expression and inflammatory cytokine production. *Foxp3* enhancer CNS2 is shown as an example of *de novo* DNA methylation during this process.

**Figure 4 biomolecules-12-01282-f004:**
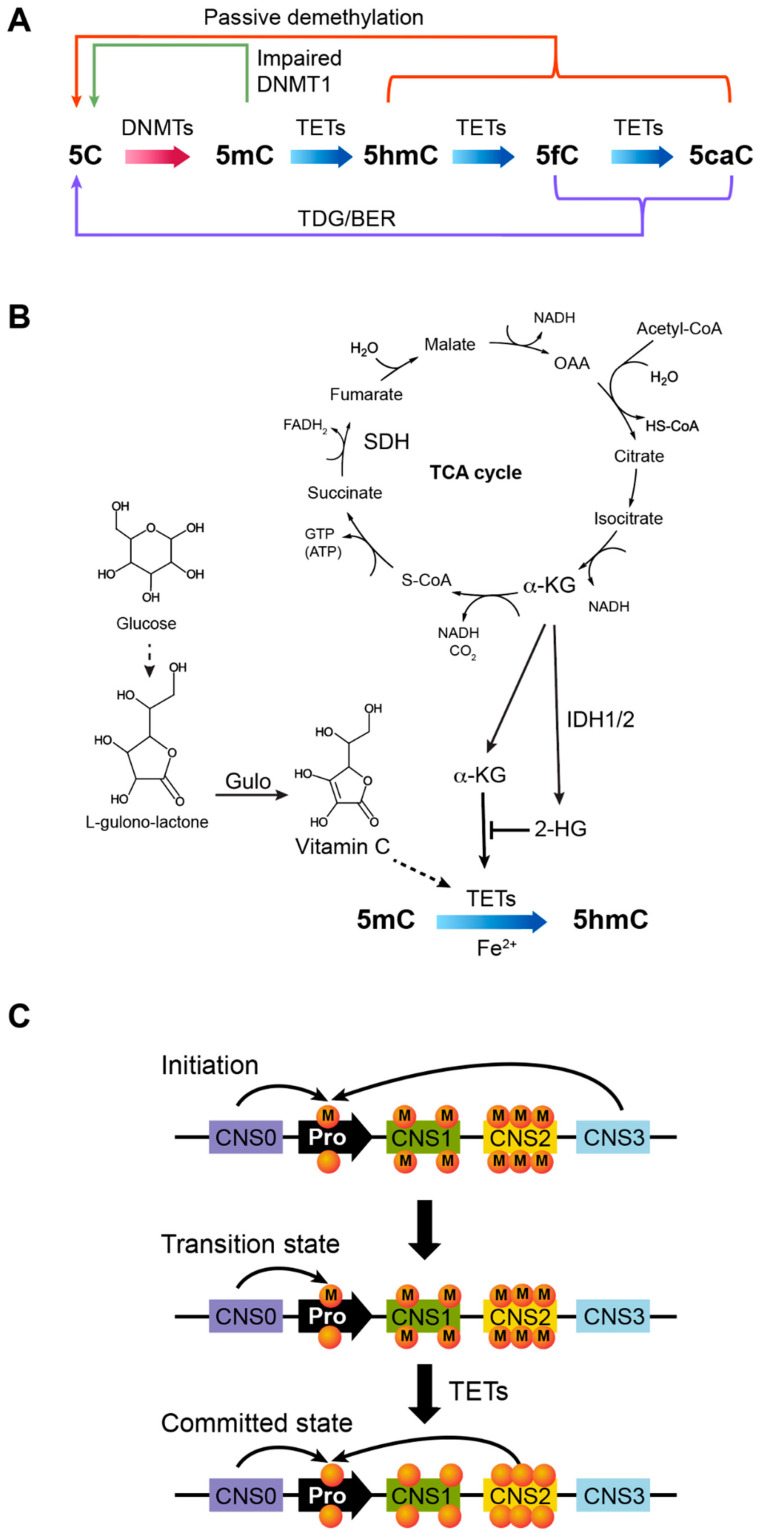
DNA demethylation by TET enzymes stabilizes Foxp3 transcription. (**A**) Schematic of the processes of TET-dependent DNA demethylation. TET enzymes catalyze sequential oxidization of 5mC to 5hmC, 5fC, and 5caC. These oxidized forms can either be diluted out during cell division or actively replaced by TDG/BER pathways. Notably, impaired DNMT1 function also results in passive DNA demethylation due to DNA replication and cell division. (**B**) TET enzymatic activity relies on vitamin C, α-KG, and Fe^2+^. For simplicity, only the first reaction from 5mC to 5hmC is shown. Vitamin C synthesis from glucose in rodents requires rate-limiting enzyme L-Gulonolactone oxidase (Gulo). α-KG is an intermediate of the TCA cycle. It is converted to 2-HG by mutant IDH1/IDH2 to inhibit TET function. (**C**) Mechanical steps involved in Foxp3 induction and maintenance. At the initiation phase, enhancers CNS0 and CNS3 coordinate to induce Foxp3 expression in precursor cells in a DNA methylation-independent manner. Upon induction, CNS0 plays a crucial role in sustaining Foxp3 expression at the transition state when TET enzymes start to demethylate the CpG sites at *Foxp3* promoter, CNS1, and CNS2. Once these CpG sites are completely demethylated, Foxp3 expression is stabilized and Treg cells are fully committed. At this stage, CNS0 and CNS2 coordinate to maintain Foxp3 transcription via DNA demethylation-independent and -dependent mechanisms, respectively.

**Figure 5 biomolecules-12-01282-f005:**
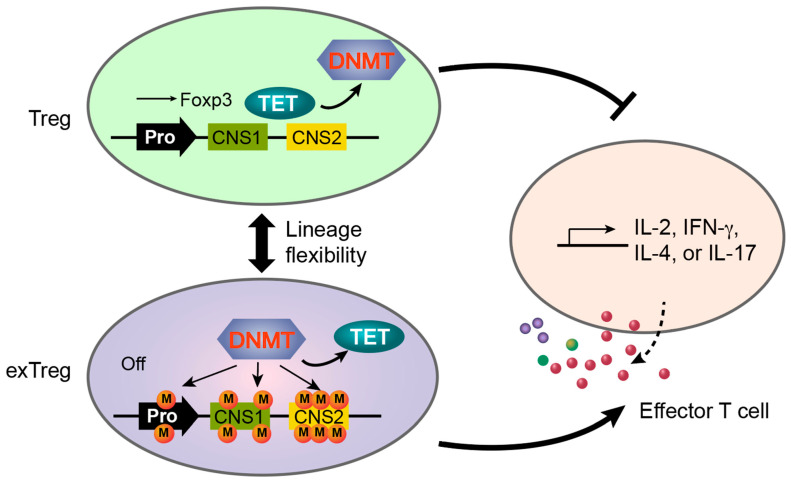
Regulation of Treg cell fate by the competition between DNMT and TET enzymes. A hypothetical model of Treg cell fate maintenance and transdifferentiation. Treg cell fate and suppressive function are maintained by TET enzymes after lineage commitment. In the absence of *TET* genes or upon stimulation by pro-inflammatory cytokines, DNMT3A/DNMT3B remethylate *Foxp3 cis*-regulatory elements (e.g., CNS1 and CNS2) and other TSDRs (not shown), leading to the transdifferentiation of Treg cells to effector-like exTreg cells characterized by silenced Foxp3 expression and pro-inflammatory cytokine production. Because DNMT and TET enzymes are targeted by distinct signaling pathways, this dynamic regulation of Treg cell fate and reprogramming fine-tunes immune tolerance accordingly to specific environmental cues likely acting in an antigen-specific manner.

## Data Availability

Not applicable.

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
