# Peer review of "DNA Methylation in Regulatory T Cell Differentiation and Function: Challenges and Opportunities"

_biomolecules, 2022, doi:10.3390/biom12091282_

Round 1

Reviewer 1 Report

This is a well-written comprehensive review of the role of DNA methylation in the regulation of T cell differentiation.  There are a few minor corrections.  One is one page 5, one on page 10, and one on page 14. 

On page 5 paragraph after 2.3 line 5 should read: DNMT1-"deficient"

On page 10 the second complete paragraph should read: Foxp3 enhancer CNS2 could potentially offer "a"

On page 14 the second paragraph near the botton line 14 should read: It would be important to "know"

Reviewer 2 Report

In the current manuscript, Bai et al. performed a well-detailed revision about the role of DNA methylation in the differentiation and function of regulatory T (Treg) cells. In particular, the authors focus on transcriptional regulation of Foxp3 expression, which plays a pivotal role in Treg cells differentiation.

Overall the Review is well written and of high interest. The role of methylation and demethylation in the regulation of Treg cells has been meticulously addressed and the manuscript will be useful for both experienced and novice researchers who are starting to emerge in this field.

I have only minor suggestions for the authors.

-Animal models mentioned in the text should be better described. For example, how Foxp3Cre did induce deletion of Dnmt1 (page 5, line 11 from the bottom)? A better description can be benefits for the readers

Minor comments.

There are some minor typos thorough the manuscript that should be checked. Following some examples:

-In figure 2 “Shortened liftespan” should be replaced with “Shortened lifespan”.

-In figure 2 caption, “to binds” should be replaced with “to bind”.

-Page 5, line 7 from the bottom: “Dnmt1-dificient” should be replaced with “Dnmt1-deficient”
